# In Vitro Study of Zirconia Surface Modification for Dental Implants by Atomic Layer Deposition

**DOI:** 10.3390/ijms241210101

**Published:** 2023-06-14

**Authors:** Tatsuhide Hayashi, Masaki Asakura, Shin Koie, Shogo Hasegawa, Akimichi Mieki, Koki Aimu, Tatsushi Kawai

**Affiliations:** 1Department of Dental Materials Science, Aichi Gakuin University School of Dentistry, 1-00 Kusumoto-cho, Chikusa-ku, Nagoya 464-8650, Japan; masaki@dpc.agu.ac.jp (M.A.); akimichi.mieki@gmail.com (A.M.); skoidats@gmail.com (K.A.); kawaita@dpc.agu.ac.jp (T.K.); 2Department of Maxillofacial Surgery, Aichi Gakuin University School of Dentistry, 2-11 Suemori-dori, Chikusa-ku, Nagoya 464-8651, Japan; koie.s824@gmail.com (S.K.); shogo@dpc.agu.ac.jp (S.H.)

**Keywords:** atomic layer deposition, zirconia, dental implants, surface modification, cell proliferation

## Abstract

Zirconia is a promising material for dental implants; however, an appropriate surface modification procedure has not yet been identified. Atomic layer deposition (ALD) is a nanotechnology that deposits thin films of metal oxides or metals on materials. The aim of this study was to deposit thin films of titanium dioxide (TiO_2_), aluminum oxide (Al_2_O_3_), silicon dioxide (SiO_2_), and zinc oxide (ZnO) on zirconia disks (ZR-Ti, ZR-Al, ZR-Si, and ZR-Zn, respectively) using ALD and evaluate the cell proliferation abilities of mouse fibroblasts (L929) and mouse osteoblastic cells (MC3T3-E1) on each sample. Zirconia disks (ZR; diameter 10 mm) were fabricated using a computer-aided design/computer-aided manufacturing system. Following the ALD of TiO_2_, Al_2_O_3_, SiO_2_, or ZnO thin film, the thin-film thickness, elemental distribution, contact angle, adhesion strength, and elemental elution were determined. The L929 and MC3T3-E1 cell proliferation and morphologies on each sample were observed on days 1, 3, and 5 (L929) and days 1, 4, and 7 (MC3T3-E1). The ZR-Ti, ZR-Al, ZR-Si, and ZR-Zn thin-film thicknesses were 41.97, 42.36, 62.50, and 61.11 nm, respectively, and their average adhesion strengths were 163.5, 140.9, 157.3, and 161.6 mN, respectively. The contact angle on ZR-Si was significantly lower than that on all the other specimens. The eluted Zr, Ti, and Al amounts were below the detection limits, whereas the total Si and Zn elution amounts over two weeks were 0.019 and 0.695 ppm, respectively. For both L929 and MC3T3-E1, the cell numbers increased over time on ZR, ZR-Ti, ZR-Al, and ZR-Si. Particularly, cell proliferation in ZR-Ti exceeded that in the other samples. These results suggest that ALD application to zirconia, particularly for TiO_2_ deposition, could be a new surface modification procedure for zirconia dental implants.

## 1. Introduction

Pure titanium (Ti) and certain Ti alloys are the most commonly used materials for dental implants because the properties of Ti include osseointegration potential, excellent biocompatibility, non-toxicity, and non-inflammation [1]. To enhance the osseointegration potential of Ti dental implants, various surface roughness and surface modification techniques are applied, which include sandblasting with large grit and acid-etched (SLA) surface treatments; hydroxyapatite coatings; anodization; and plasma spray coatings [2,3,4,5,6,7]. However, although cases are rare, Ti allergy has been clinically reported [8,9], and the gray color of Ti sometimes becomes an aesthetic problem, particularly in the thin gingival mucosa. For these reasons, zirconia dental implants have recently been clinically applied as alternatives to Ti dental implants, mainly in Europe. In particular, yttria-stabilized tetragonal zirconia polycrystal (Y-TZP) has been clinically applied as a representative material for ceramic dental implants because of its exceptional mechanical properties, such as its sufficient fracture toughness, high flexural strength, and adequate Young’s modulus, as well as excellent biocompatibility, low sensitivity to dental plaque formation, and white color [10].

However, a polished zirconia surface exhibits hydrophobicity and is bio-inert. As hydrophilicity is generally required for the integration of zirconia dental implants into the alveolar bone, surface modification of zirconia dental implants is performed to obtain high osseointegration potential, as for Ti dental implants [11,12]. For instance, sandblasting, etching, laser irradiation, and sintering are used to roughen zirconia surfaces [13,14,15,16], and ultraviolet and plasma irradiations are used to activate these surfaces [17,18]. In addition, surface-activated coating treatments have been applied to zirconia, including glass and apatite coatings [19,20]. Although various surface modification procedures have been devised, an appropriate standard procedure is yet to be established for zirconia.

Atomic layer deposition (ALD) is a vapor-phase nanotechnology-based technique in which ultra-thin or thin films of metals, metal oxides, sulfides, and nitrides are deposited on various materials. Using ALD, high-quality thin films can be deposited via a routine atomic layer-by-layer procedure, and the film thickness can be controlled on complex three-dimensional materials, as well as on materials with high aspect ratios or porous surfaces. Moreover, thin films deposited using ALD appear uniform across the material, conformal, and pinhole-free [21]. Accordingly, ALD can be effectively used to modify the surface chemistry and functionality of engineering-related and biologically important surfaces. Furthermore, it can also be used to enhance the chemical, mechanical, electrical, and other properties of materials used in biomedical engineering and biological sciences [22]. Furthermore, several studies in the biomedical field have evaluated osteoblast viability and inhibiting bacterial adhesion on thin film deposited Ti using ALD [23,24]; however, similar experiments on zirconia have been rarely reported.

In this present study, titanium dioxide (TiO_2_; ZR-Ti), aluminum oxide (Al_2_O_3_; ZR-Al), silicon dioxide (SiO_2_; ZR-Si), and zinc oxide (ZnO; ZR-Zn) thin films were deposited on zirconia discs (ZR) using ALD for zirconia surface modification, and the proliferation abilities of mouse fibroblasts (L929) and mouse osteoblastic cells (MC3T3-E1) on each sample were evaluated.

## 2. Results

### 2.1. Scanning Transmission Electron Microscopy Observation of a Thin Film on ZR and Measurement of the Thin-Film Thickness

Figure 1A-1–D-1 presents the cross-sectional scanning transmission electron microscopy (STEM) images of ZR-Ti, ZR-Al, ZR-Si, and ZR-Zn, and Figure 1A-2–D-2 presents the synchronized elemental images of Zr, O, and the target elements in each sample. These results indicate that TiO_2_, Al_2_O_3_, SiO_2_, and ZnO films were directly deposited on the zirconia surfaces in each case, with thicknesses of 41.67, 42.36, 62.50, and 61.11 nm, respectively.

### 2.2. Contact Angle

The wettability test results are shown in Figure 2. Figure 2A presents the side-view image of a 0.5 μL ultra-pure water droplet placed on each specimen. The average contact angles of the ultra-pure water droplet on ZR, ZR-Ti, ZR-Al, ZR-Si, and ZR-Zn were 86.9, 80.5, 91.5, 61.6, and 98.5°, respectively. The ZR and ZR-Ti surfaces presented slight hydrophilicity, whereas ZR-Al and ZR-Zn demonstrated slight hydrophobicity. The contact angle on the ZR-Si surface was significantly lower than that on all the other specimens (Figure 2B).

### 2.3. Adhesion Strength

The scratch test results are shown in Figure 3. Figure 3A shows a representative sample surface (ZR-Ti) after an adhesion test. The yellow line indicates the area where a change in the sensor output signal was detected, and the small red circle indicates the adhesion strength measurement point. The average adhesion strengths of ZR-Ti, ZR-Al, ZR-Si, and ZR-Zn were 163.5, 140.9, 157.3, and 161.6 mN, respectively, with no statistical significance among the samples (Figure 3B).

### 2.4. Eluted Element Measurements

Figure 4 shows the immersion test results for ZR, ZR-Ti, ZR-Al, ZR-Si, and ZR- Zn. For all the samples, the Zr content was below the detection limit. Additionally, the Ti and Al elution amounts in ZR-Ti and ZR-Al were below the detection limits. This indicates that the thin films of TiO_2_ and Al_2_O_3_ on zirconia were chemically stable. By contrast, Si was detected in ZR-Si, with concentrations of 0.018 and 0.019 ppm in the first and second weeks, respectively. Furthermore, Zn was prominently detected in the ZR-Zn sample, with concentrations of 0.611 and 0.695 ppm in the first and second weeks, respectively.

### 2.5. Cell Proliferation and Cell Morphology

L929 and MC3T3-E1 cell proliferation was analyzed throughout the culture period and the results are shown in Figure 5A,B, respectively. For L929, statistically significant differences among samples were observed on day 3 only; however, the number of L929 cells gradually increased over time in ZR, ZR-Ti, ZR-Al, and ZR-Si and reached confluence on day 5. For the MC3T3-E1 cell culture, significant differences among the samples were apparent during each culture period. The number of cells on ZR, ZR-Ti, and ZR-Al increased dramatically from day 1 to day 4 and approached confluence on day 4; over-confluence was observed on day 7. The cell proliferation in ZR-Si was slower than that in the other samples throughout the culture period.

SEM images of typical L929 and MC3T3-E1 cell morphologies on ZR, ZR-Ti, ZR-Al, ZR-Si, and ZR-Zn are shown in Figure 5C,D, respectively. The L929 cell morphologies on ZR, ZR-Ti, ZR-Al, and ZR-Si were spindle-shaped and polygonal, and the cytoskeletons were well-developed from day 1. The MC3T3-E1 cell morphologies on ZR, ZR-Ti, ZR-Al, and ZR-Si were similar to those of L929 cells. Both cell lines presented paving stone-like morphologies upon confluence.

By contrast, for both the L929 and MC3T3-E1 cell cultures on ZR-Zn, the number of cells did not increase from day 1, and confluence was not reached. Moreover, only a few, irregularly shaped cells were observed, as apparent from the SEM images.

## 3. Discussion

ALD is a relatively new technique that is gradually being applied to the field of nanotechnology, bio-nanotechnology, and ultra-thin/thin film deposition; this technique is ideal for film deposition on the nanometer or Ångström length scales. ALD is considered a promising approach for controlling the hydrophilicity/hydrophobicity characteristics of biomedical surfaces, depositing conformal ultra-thin coatings with desirable properties on biomedical materials with high aspect ratios, controlling the antibacterial properties of material surfaces, and developing multifunctional biomaterials for medical implants and other medical devices [25,26,27].

The ALD thin-film deposition process is based on the repetition of four cycles. First, a precursor, which is usually a metallic element surrounded by organic functional groups, is introduced to a reactor and chemically absorbed into the target material. The unreacted precursor is then purged from the reaction chamber using an inert gas, such as N_2_ or Ar. Next, an oxidizing agent, such as O_2_, O_3_, or water vapor, is pulsed into the reactor and it reacts with the precursor absorbed on the material surface. Finally, the unreacted oxidizing agent and surface reaction by-products are purged from the reaction chamber using inert gas. After each cycle, a monolayer of the desired substance (with atomic thickness) is deposited on the surface. These four steps are repeated cyclically to deposit a thin film with the desired thickness. Therefore, thin-film thickness control with atomic-level precision is expected when ALD is employed [22].

In this study, TiO_2_, Al_2_O_3_, SiO_2_, and ZnO deposition onto ZR was confirmed using STEM and energy-dispersive X-ray spectroscopy (EDS) analyses. During the wettability test, the contact angle of ultra-pure water on ZR-Si was significantly lower than that on the other specimens. The ZR and ZR-Ti surfaces were found to be slightly hydrophilic, whereas the ZR-Al and ZR-Zn surfaces were found to be slightly hydrophobic. Moreover, L929 cell proliferation increased over time on ZR, used as a control, and on ZR-Ti, ZR-Al, and ZR-Si, indicating good cell compatibility and well-developed cytoskeletons. In the MC3T3-E1 cell culture, cell proliferation on ZR-Si was slower than that on ZR, ZR-Ti, and ZR-Al. Although there were almost no statistically significant differences, cell proliferation on ZR-Ti was notably higher than that on ZR and ZR-Al. In addition, the MC3T3-E1 cell compatibility appears to have surpassed that of the L929 cells, as the former reached confluence on day 4 of culturing, compared to day 5 for the latter. Although the exact mechanism is unknown and may depend on the material and cell type, the hydrophilicity of a material surface has been reported to increase cell adhesion and proliferation [4,15,17,20]. However, the hydrophilicity/hydrophobicity of the specimen surface did not seem to have a significant effect on the results of this cell proliferation test.

As reported previously, nanostructured TiO_2_ film deposited on Ti6Al4V [28], bioactive TiO_2_ film deposited on polyetheretherketone [29], Al_2_O_3_ thin film deposited on glass cover slips [30], one-dimensional Al_2_O_3_ nanostructures deposited on round-shaped glass cover slips [31], and TiO_2_-SiO_2_ (30:70) coating produced on Ti disks [32] have all demonstrated good biocompatibility [28,29,30,31,32], and our experimental results were approximately consistent with these reports. A reason for the good proliferation of L929 and MC3T3-E1 cells on ZR-Ti and ZR-Al is that in the immersion test, the elution amounts of Ti and Al from both samples were below the detection limits, i.e., both TiO_2_ and Al_2_O_3_ depositions were chemically stable on ZR. The gradual proliferation of MC3T3-E1 cells on ZR-Si was probably attributed to the slight elution of Si, i.e., the SiO_2_ thin-film-depositing state was chemically somewhat unstable. By contrast, for both cell cultures, the number of cells on ZR-Zn did not increase from the first day, and irregular cell morphologies were observed.

The target thin-film thickness was set to 50 nm but an error of approximately ±10 nm was observed. This is because each molecule differed in size and, thus, the number of deposition cycles varied with the deposition molecule. Therefore, it is difficult to specify the number of cycles necessary to obtain a target thin-film thickness. Furthermore, it is currently unknown whether the differences in the thin-film thickness affected the element elution, adhesion strength, or cell proliferation results. Hence, more precise deposition conditions must be determined to obtain the target thin-film thickness for each deposition molecule and examine if the differences in thin-film thickness affect the abovementioned outcomes.

The dental implant insertion torque is an indicator of the stability of a primary implant and reflects the cutting resistance of the bone at the osteotomy site during dental implant placement, which is expressed in Newton centimeters (Ncm). An insertion torque value of ≥32 Ncm is recommended to secure primary stability, particularly when immediate implant loading is assumed [33,34]. Therefore, there is concern that a thin film applied for surface modification could delaminate from the implant body during implant placement. Accordingly, here, an ultra-thin-film scratch tester was used to measure the adhesion strength of each deposited thin-film sample. A scratch tester is characterized by horizontal excitation of the stylus, which was made from diamond in this case. In this test, the load applied to the stylus is gradually increased, causing wear on the thin-film surface, which eventually causes film breakdown and delamination. The stylus-tip state of vibration changes during this process; thus, the point at which film delamination occurs can be identified from the change in the sensor output signal, and the load value applied at the point of delamination can be measured [35,36]. In this study, the average adhesion strength of each sample was approximately 155 mN; however, the correlation with the recommended placement torque for dental implant placement (≥32 Ncm) is unknown currently. This aspect must be verified in future studies.

In a previous study, ZrO_2_, SiO_2_, and ZnO thin films were deposited on Ti using ALD [37]. The eluted Si amount was less than the detection limit; however, in this study, 0.019 ppm of Si was eluted after 2 weeks of immersion. This elution occurred even though the thin films were deposited under the same conditions in both studies. Therefore, the ALD outcome may depend on the characteristics and surface roughness of the substrate material. Furthermore, Zn elution from ZR-ZnO was observed to be remarkable; this was also observed when ZnO was deposited on Ti [37]. One possible explanation for why the L929 and MC3T3-E1 cell numbers did not increase on ZR-Zn was an extreme change in the pH of the ZR-Zn culture medium owing to the elution of Zn from the ZR-Zn. However, the medium was alkalescent at pH 7.5–7.8; as for the other samples, the total amount of Zn eluted during two weeks of immersion was very small (0.695 ppm). Therefore, the eluted Zn was not assumed to have altered the pH of the culture medium. Although it is not clear why neither cell type proliferated on ZR-Zn in our previous [37] and the present study, the poor proliferation of human gingival fibroblasts on ZnO films formed on Ti compared to that on Ti has been reported [38]. By contrast, eluted Zn^2+^ has been reported to increase the viability of epithelial cells on ZnO-deposited zirconia [39], increasing cell adhesion, migration, proliferation, and osteogenic differentiation of MC3T3-E1 on ZnO-deposited sandblasted acid-etched zirconia [40] and enhancing the osteoblast differentiation of human bone-marrow-derived mesenchymal cells on Zn-modified Ti [41]. In addition, micro- and nanostructured TiO_2_/ZnO coatings on Ti have been found to promote SaOS-2 osteoblastic cell adhesion and differentiation [42]. Thus, the results of our studies challenge the results published in previous reports, warranting further experimental validation.

Ti is a highly oxidizable metal, and a TiO_2_ layer can be formed immediately upon exposure to the atmosphere [43]. In fact, in a previous study, STEM and EDS analyses of Ti samples after thin-film deposition using ALD confirmed the spontaneous formation of an approximately 10–15 nm thick TiO_2_ layer between Ti and the deposited thin film [37]. This implies that when a Ti dental implant is implanted, it is not Ti that is in direct contact with the bone but TiO_2_. In other words, the TiO_2_ that naturally forms on the Ti surface is osseointegrated. Therefore, if a TiO_2_ thin film is deposited on the surface of a zirconia dental implant using ALD, higher osseointegration ability than that of a conventional zirconia dental implant can be achieved. Furthermore, because zirconia is a ceramic, the problems of corrosion and allergy concerning Ti can be eliminated, and dental implants with high strength and excellent biocompatibility can be developed.

This study had some limitations. As we decided to only focus on ALD on zirconia, we only studied a singular technique. Further destructive surface treatments have been introduced including ultra-short laser pulses to the zirconia surface, where the surface roughness increases with a minor tetragonic to monoclinic transformation [44,45]. Future studies could investigate the combination of an additive (such as ALD) and destructive techniques (such as laser treatment). Furthermore, we assumed that cell viability had little influence on the thin film; however, to assess the influence of cell viability on the thin film, further analysis of the samples should be performed after cell culture, such as how the thin films have degraded, changes in mechanical strength, and changes in chemistry. Additionally, we did not assess the effects of zirconium and oxygen present within the zirconia sample or the crystal structure of zirconia on thin film deposition. Future studies should vary these parameters to assess the effects they may have.

## 4. Materials and Methods

### 4.1. Preparation of Zirconia Samples

A commercially available zirconia block (Ceramill ZOLID 71L (20 mm) zirconium dioxide (ZrO_2_); Amman Girrbach AG, Vorarlberg, Austria) was used in this study. First, a computer-aided design/computer-aided manufacturing (CAD/CAM) system (Ceramill^®^ mind/ Ceramill^®^ motion 2; Amman Girrbach AG, Vorarlberg, Austria) was used to machine the zirconia block into a cylindrical shape (height: 20 mm; diameter: 12.6 mm). Disk-shaped samples (thickness: 630 μm) were then cut using a low-speed diamond saw (Isomet LS; Buehler, Lake Bluff, IL, USA), and the cut samples were sintered in an electric furnace (Type 51314; KOYO LINDBERG LTD., Nara, Japan). The temperature of the electric furnace was increased by 300 °C/h until it approached 1450 °C, after which it was maintained at 1450 °C for 2 h and then cooled down for 6 h. The final sample dimensions were approximately 500-micrometer thickness and 10-mm diameter. The sintered zirconia samples were polished using a barrel polishing machine (TWIN BARREL; DCL TANIMOTO Co., LTD., Hyogo, Japan) and further sequentially polished with abrasives down to a 0.3-micrometer alumina suspension using a rotary polishing machine (Ecomet III; Buehler, Lake Bluff, IL, USA). This polishing step was followed by washing with a synthetic detergent, ultrasonic cleaning for 20 min, and air drying.

### 4.2. ALD on ZR

Prior to the ALD treatment, the ZR were irradiated with plasma for 10 min to maximally remove elemental contaminants. TiO₂, Al_2_O_3_, SiO₂, and ZnO thin films were deposited on ZR using an ALD reactor (At-400; ANRIC TECHNOLOGIES, Lexington, MA, USA). The TiO_2_, Al_2_O_3_, SiO_2_, and ZnO thin films were deposited using tetrakis(dimethylamido)titanium (TDMATi; Oakwood Products, Inc., West Columbia, SC, USA), trimethylaluminum (TMA; Gas-Phase Growth Ltd., Tokyo, Japan), tris(dimethylamino)silane (TDMASi; Oakwood Products, Inc., West Columbia, SC, USA), and diethylzinc (DEZ; Oakwood Products, Inc., West Columbia, SC, USA), respectively, as precursors. With regard to oxidizing agents, ultra-pure water (Grade II) was used for TiO_2_, Al_2_O_3_, and ZnO thin films, and O_3_ was used for the SiO_2_ thin film. The target deposition thickness was set to 50 nm. Details of the deposition conditions are listed in Table 1. After the ALD treatment, the samples were sterilized with ethylene oxide gas before being employed in the cell culture experiments.

### 4.3. STEM Observation

The thickness and elemental distribution of each thin film were determined using a STEM equipped with an EDS (JEM-2100F; JEOL Ltd., Tokyo, Japan). STEM parameters used were as follows: electron-beam spot size, 1 nm; sample acceleration, 200 kV.

### 4.4. Wettability Test

The hydrophilic or hydrophobic property of each specimen surface was evaluated using an automatic contact angle meter (DMs-400; Kyowa Interface Science, Co., Ltd., Saitama, Japan) to measure the contact angle between a 0.5 μL drop of ultra-pure water and each specimen surface. Based on the obtained image of a sessile drop, the contact angle meter measured the angle between the point of intersection of the drop profile and the projection of the specimen surface.

### 4.5. Scratch Test

A scratch test was performed using an oscillating microscratch tester (CSR5100; RHESCA Co., Ltd., Tokyo, Japan) to measure the ZR-Ti, ZR-Al, ZR-Si, and ZR-Zn adhesion strengths to ZR. Each sample was analyzed three times, and the average was calculated. The change point on the scratch mark of each sample was observed with a laser microscope (LEXT OLS4000; Olympus Corporation, Tokyo, Japan).

### 4.6. Immersion Test

Each sample was immersed in 50 mL conical tubes containing 45 mL of ultra-pure water. The tubes were placed in a constant-temperature oven (DKM400; Yamato Scientific Co., Ltd., Tokyo, Japan) set at 37 °C and stored statically for two weeks. After storage, the concentrations of the eluted elements in the immersed solution were measured using an inductively coupled plasma (ICP) atomic emission spectrometer (Optima 7300 DV; PerkinElmer Japan Co., Ltd., Yokohama, Japan).

### 4.7. Cell Culture and Cell Proliferation Assay

The procedures used to obtain cell concentrations and cell cultures were based on protocols established in previous studies [37,46,47]. 

In this study, mouse fibroblasts (L929, NCTC clone 929, connective tissue, mouse; DS Pharma Biomedical Co., Osaka, Japan) and osteoblastic cells (MC3T3-E1, calvaria, mouse; DS Pharma Biomedical Co., Osaka, Japan) were used for the cell culture experiments. The L929 cells were cultured at 37 °C in Eagle’s minimal essential medium (MEM; Life Technologies Japan Ltd., Tokyo, Japan) supplemented with 5% fetal bovine serum (FBS; Life Technologies Japan Ltd., Tokyo, Japan), penicillin (100 IU/mL; Life Technologies Japan Ltd., Tokyo, Japan), and streptomycin (100 µg/mL; Life Technologies Japan Ltd., Tokyo, Japan) in a humidified atmosphere with 5% CO_2_. The MC3T3-E1 cells were cultured under the same conditions in alpha-Eagle’s MEM (Life Technologies Japan Ltd., Tokyo, Japan) supplemented with 10% FBS, penicillin (100 IU/mL), and streptomycin (100 µg/mL).

Each sample was then placed in a 48-well microplate. The cells were suspended at a density of 1 × 10^5^ cells/mL in their respective medium and seeded at 200 µL/well. The cells were also seeded directly on ZR (without ALD treatment) as a control. The culture media were changed every other day. The L929 cells were cultured for 5 days, and cell proliferation was measured from three samples on days 1, 3, and 5. MC3T3-E1 cells were cultured for 7 days, and cell proliferation measurements were performed on three samples on days 1, 4, and 7. The cell proliferation at each time point was measured via a water-soluble tetrazolium salt (WST) assay using Cell Counting Kit-8 (Dojindo Laboratories, Kumamoto, Japan; optical density (O.D.) at 450 nm).

### 4.8. pH Measurement of Culture Medium

During the culture period, the pH values of both L929 and MC3T3-E1 culture media were measured using a pH meter (AS-pH-11; HORIBA Advanced Techno, Co., Ltd., Kyoto, Japan).

### 4.9. Cell Morphology Observation

To observe the post-culture morphologies, the L929 and MC3T3-E1 cell samples were fixed with 2% glutaraldehyde (TAAB Laboratories Equipment Ltd., Aldermaston, UK) for 24 h, followed by incubation in 1% osmium tetroxide (TAAB Laboratories Equipment Ltd., Aldermaston, UK) for 2 h, sequential dehydration in increasing alcohol concentrations, and freeze-drying. Finally, the samples were sputter coated with platinum and imaged using a SEM (JXA-iH200F; JEOL Ltd., Tokyo, Japan) at 10 kV acceleration. The L929 cell morphology was observed on days 1, 3, and 5, and that of the MC3T3-E1 cells was observed on days 1, 4, and 7.

### 4.10. Statistical Analysis

In this study, all quantitative data were collected as the mean ± standard deviation (SD) (*n* = 3) values. Statistical analyses for the wettability test, scratch test, and cell proliferation assay were performed using one-factor analyses of variance, followed by Tukey’s post-hoc tests. *p* values of less than 0.05 were considered statistically significant.

## 5. Conclusions

In conclusion, the deposition of TiO_2_, Al_2_O_3_, SiO_2_, and ZnO thin films onto ZR using ALD provided promising results for dental implant creation. The number of L929 and MC3T3-E1 cells on the ZR-Ti, ZR-Al, and ZR-Si increased over time, and, in particular, cell proliferation was most prominent on ZR-Ti, demonstrating excellent biocompatibility. We found that the contact angle on the ZR-Si surface was significantly lower than the other specimens; however, there were no statistically significant differences in the thin-film adhesion strength among the samples and wettability did not seem to have a significant effect on the results of this cell proliferation test. Additionally, as the TiO_2_ and Al_2_O_3_ deposited on the ZR were chemically stable, the amounts of Ti and Al elution were below detection limits in the immersion test. These results indicate that the use of ALD for coating zirconia dental implants, particularly for TiO_2_ deposition, could be employed as a novel surface modification technique. Future studies should assess the effect of zirconium and oxygen within the zirconia sample and the crystal structure of zirconia on thin film deposition.

## Figures and Tables

**Figure 1 ijms-24-10101-f001:**
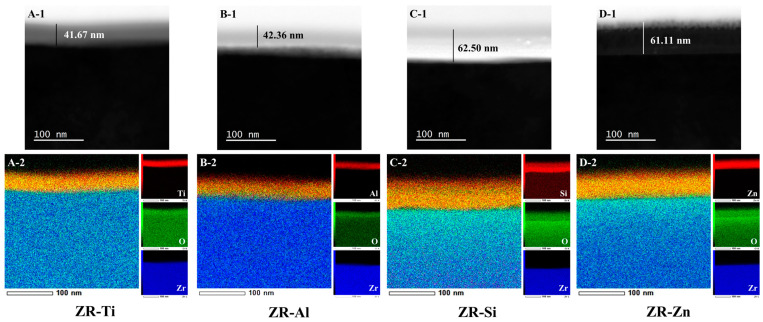
Scanning transmission electron microscope (STEM) cross-sectional images and energy dispersive spectrometry (EDS) elemental color mapping images of ZR-Ti ((**A-1**,**A-2**), respectively), ZR-Al ((**B-1**,**B-2**), respectively), ZR-Si ((**C-1**,**C-2**), respectively), and ZR-Zn ((**D-1**,**D-2**), respectively).

**Figure 2 ijms-24-10101-f002:**
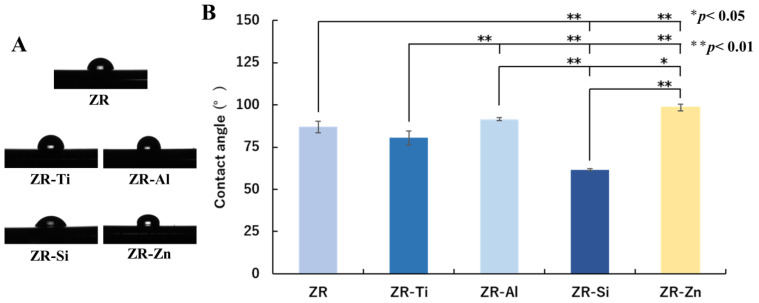
Contact angles between the ultra-pure water droplet and each specimen. (**A**) Side-view images of a 0.5 μL ultra-pure water droplet placed on each sample. (**B**) Contact angles of the ultra-pure water droplet on ZR, ZR-Ti, ZR-Al, ZR-Si, and ZR-Zn, measured using an automatic contact angle meter. The data are reported as mean ± SD values (*n* = 3).

**Figure 3 ijms-24-10101-f003:**
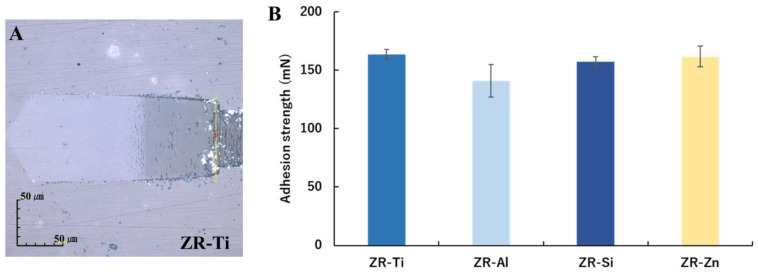
Adhesion test results for ZR-Ti, ZR-Al, ZR-Si, and ZR-Zn. (**A**) Sample surface image obtained using laser microscope after adhesion testing. The yellow line indicates the area where the change in the sensor output signal was detected. The small red circle denotes the adhesion strength measurement point. Measurements were performed three times for each sample and the average was calculated. (**B**) Adhesion strength values for ZR-Ti, ZR-Al, ZR-Si, and ZR-Zn, obtained using an oscillating microscratch tester. The data are reported as mean ± SD values (*n* = 3).

**Figure 4 ijms-24-10101-f004:**
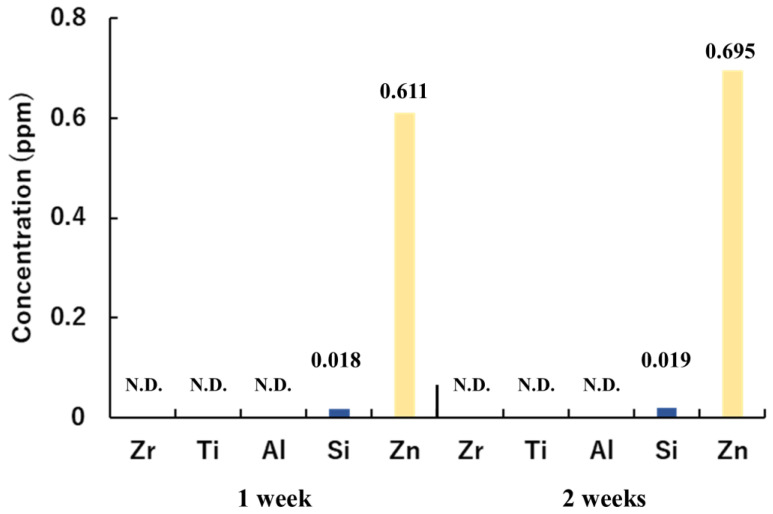
Immersion test results obtained using inductively coupled plasma (ICP) atomic emission spectrometry for ZR, ZR-Ti, ZR-Al, ZR-Si, and ZR- Zn. The graph shows the elemental elution concentrations of Zr, Ti, Al, Si, and Zn. (N.D.; Not detected).

**Figure 5 ijms-24-10101-f005:**
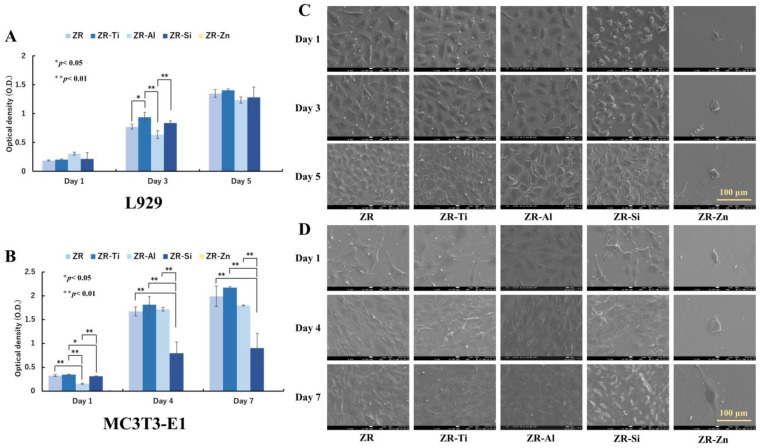
(**A**,**B**) L929 and MC3T3-E1 cell proliferation, respectively, on ZR, ZR-Ti, ZR-Al, ZR-Si, and ZR-Zn. The data are reported as mean ± SD values (*n* = 3). (**C**,**D**) Scanning electron microscope (SEM) images of L929 (days 1, 3, and 5 of culturing) and MC3T3-E1 (days 1, 4, and 7 of culturing) cells, respectively, for ZR, ZR-Ti, ZR-Al, ZR-Si, and ZR- Zn.

**Table 1 ijms-24-10101-t001:** ALD thin-film deposition conditions.

Thin Films	Precursor/Pre-Heating Temp. (°C)	Pulse (s) × Times	N_2_ Purge (s)	Cycles (Times)	Deposition Temp. (°C)
	Oxidizing Agent				
TiO_2_	TDMATi/83	2.35 × 3	8	667	180
	H_2_O	2.35 × 2	10		
Al_2_O_3_	TMA/RT	2.35 × 3	11	300	150
	H_2_O	2.35 × 2	13		
SiO_2_	TDMASi/54	2.35 × 3	12	536	175
	O_3_	2.35 × 2	10		
ZnO	DEZ/RT	2.35 × 2	8	300	175
	H_2_O	2.35 × 2	10		

(RT: room temperature).

## Data Availability

The data presented in this study are available on request from the corresponding author. The data are not publicly available due to its confidentiality.

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
