# Peer review of "In Vitro Study of Zirconia Surface Modification for Dental Implants by Atomic Layer Deposition"

_ijms, 2023, doi:10.3390/ijms241210101_

Round 1

Reviewer 1 Report (New Reviewer)

The paper "Surface Modification of Zirconia for Dental Implants by Atomic Layer Deposition; An in vitro study" presents an in vitro study for some dental implants. The authors have performed an interesting study, but they need to improve some minor aspects:

1. I suggest changing the title to: In vitro study of zirconia coatings for dental implants by atomic layer deposition."

2. The introduction needs some extra information regarding other types of zirconia coating techniques and also some data on the influence of Zr and O in the base material. Suggested references: 10.37358/rc.19.6.7321; 10.1016/j.apsusc.2015.05.111.

3. In Section 2.1, please add some SEM/TEM parameters.

4. The scale bars of figures 5C and D are not clear.

5. Materials and methods after discussion? Please explain.

6. Conclusions are too short. Add more experimental data.

7. In your general opinion, please explain more about cell viability and see how it influences the coating.

The rest is fine.

Author Response

Reviewer 2 Report (New Reviewer)

Very good idea for an additive technique is the suface of zirconia.

Please check the mistakes in number of Figures line 109,124,146

I' ll propose a small paragraph in the discussion describing other destructive surface treatment that have been introduced. As an example ultra short laser pulses in zirconia surface (check 1. Delgado-Ruíz RA, Calvo-Guirado JL, Moreno P, et al. Femtosecond laser microstructuring of zirconia dental implants. J Biomed Mater Res B Appl Biomater. 2011;96(1):91-100. doi:10.1002/jbm.b.31743 and 2.Tzanakakis E, Kontonasaki E, Voyiatzis G, Andrikopoulos K, Tzoutzas I. Surface characterization of monolithic zirconia submitted to different surface treatments applying optical interferometry and raman spectrometry. Dent Mater J. 2020;39(1):111-117. doi:10.4012/dmj.2018-358). It is very interesting that surface roughness increases with minor tetragonic to monoclinic tranformation after this laser treatment and it could a very innovative idea to combine an additive (e.g. ALD) with a destructive technique (Sandblasting, Laser treatment). 

Overall a very well presented work in the field of zirconia implants.

Very good language.

Author Response

This manuscript is a resubmission of an earlier submission. The following is a list of the peer review reports and author responses from that submission.

Round 1

Reviewer 1 Report

Abstract. This section is adequate and show the summary of the paper.

Introduction. This investigation is a paper that presents information for researchers in the field of surface of implants. Zirconia dental implants have recently been clinically applied alternatives to Ti dental implants. Zirconia implants reported  good mechanical properties, as well as excellent biocompatibility, low sensitivity to dental plaque formation, and aesthetics. Surface modification of zirconia dental implants is performed to obtain a high osseointegration potential, as sandblasting, etching, laser irradiation, and ultraviolet and plasma irradiations are used to activate the zirconia surface. Accordingly, atomic layer deposition (ALD) can be effectively used to modify the surface chemistry and functionalization of surfaces and can be used to enhance the chemical, mechanical, electrical, and other properties of materials used in applications in the fields of biological sciences.

The authors must include in this end of this section, recent in vitro studies of this surface modification with ALD on celular viability, because the aim of the study is analize in vitro results of this surface modifications about fibroblast ans osteoblast cells.

Materials and methods. This section is adequate. This study in vitro was designed to analyze the different implant surfaces behabvior with preparation of discs with ALD, scanning microscopy, cell culture and proliferation, etc).

However, only the subsection 4. 6. Cell culture and cell proliferation assay includes some references. In the rest of subsections not found references. The authors must improve this section with several references in each subsection.

Results.

The results indicate that TiO2, Al2O3, SiO2 and ZnO were directly deposited on the zirconia surface in each case. The scratch test shows a representative sample surface (ZR-Ti) after an adhesion test. The L929 and MC3T3-E1 cell proliferations were measured throughout the culture period.  For L929, cells gradually increased over time in the ZR, ZR-Ti, ZR-Al, and ZR-Si. For the MC3T3-E1 cell culture, the number of cells on the ZR, ZR-Ti, and ZR-Al increased dramatically from days 1 to day 4, with confluence occurring on day 4.

This section is correct and showes  several adequate figures.

Discussion.

The fisrst paragrpahes must includ more references of the application fo ADL in implant surfaces.

The third paragraph reported the main results of this study. But the authors not analize the significance of these results and the comparation with other experimental studies.

The fourth paragraph need be more explained by the authors.

 Conclusions. This section is not adequate according to the paper. It is very long and includes a repetition of results

Conclusively, the study is not ready for publication.

Reviewer 2 Report

The manuscript entitled “Surface Modification of Zirconia for Dental Implants by Atomic Layer Deposition; An In Vitro Study”, submitted for evaluation to Coatings, presents the evaluation of sucrose concentration effect on S. mutans adhesion to dental materials.

In general, the work is clear and written in the correct language. The structure of the manuscript is relatively clear. However, there is no significant impact of the coatings presence on the properties of coated material, in particular in relation to hydrophobicity/hydrophilicity and cytotoxicity. Among all tested coatings, the increase of cells proliferation was noted only for Zr-Ti, in comparison with bare Zr material – in addition, this effect was noted exclusively for L929 cells. So, is the obtained benefit satisfactory in the light of the need to carry out a complex four-stage modification of the material?

 COMMENTS TO AUTHORS

1.  Authors declared that “…, a polished zirconia surface exhibits hydrophobicity and is bio-inert. As hydrophilicity is required for the integration of zirconia dental implants into the alveolar bone, surface modification of zirconia dental implants is performed to obtain high osseointegration potential, as for Ti dental implants [11,12].” I believe that since this feature is supposed to be crucial for the biological properties of the modified samples, the wettability angle for the modified samples should be assessed in comparison with the control. Perhaps this measurement would reveal some more significant differences between the properties of bare and coated zirconium.

Reviewer 3 Report

This manuscript demonstrates an effective surface modification procedure using Atomic Layer Deposition (ALD) on zirconia disks. The authors selected titanium dioxide, aluminum oxide, silicon dioxide, and zinc oxide to modify the surface of zirconia for enhancing the proliferation ability of cells. Overall, the manuscript provides detailed conditions and procedures for surface modification, and the authors found that titanium dioxide-treated zirconia disks were the most suitable material. However, in order to increase the manuscript's impact and improve its clarity, it would be beneficial for the authors to provide a more detailed explanation of the mechanism behind the different effects of each deposited material on cells. Here are some specific comments.

1. It is recommended to add data on attachment after culturing for 6 hours.

2. Please discuss the specific mechanisms behind the effects of each deposited material on cell fate.

3. During implant placement, external pressure and friction are exerted. Please discuss whether the deposited condition is appropriate for implant use.

4. Please use AFM and SEM to check the structural properties of each surface.

5. Please confirm the chemical properties of each sample.

6. Please conduct differentiation into osteoblasts using Alizarin Red S staining.

There was no problems to read the manuscript.

Round 2

Reviewer 1 Report

The review of the authors is correct.

Reviewer 2 Report

none

none

Round 3

Reviewer 2 Report

OK

none
